# `HASSLE-free`: A unified framework for Sparse plus Low-Rank Matrix Decomposition for LLMs

Mehdi Makni[1,*], Kayhan Behdin[2], Zheng Xu[3], Natalia Ponomareva[3], Rahul Mazumder[1]

[1]MIT    [2]LinkedIn    [3]Google Research

The impressive capabilities of large foundation models come at a cost of substantial computing resources to serve them. Compressing these pre-trained models is of practical interest as it can democratize deploying them to the machine learning community at large by lowering the costs associated with inference. A promising compression scheme is to decompose foundation models' dense weights into a sum of sparse plus low-rank matrices. In this paper, we design a unified framework coined `HASSLE-free` for (semi-structured) sparse plus low-rank matrix decomposition of foundation models. We introduce the local layer-wise reconstruction error objective for this decomposition and demonstrate that prior work solves an approximation of this optimization problem. We provide efficient and scalable methods to obtain good solutions to the *exact* optimization program. `HASSLE-free` substantially outperforms state-of-the-art methods in terms of the layerwise reconstruction error and a wide range of LLM evaluation benchmarks. For the Llama3-8B model with a 2:4 sparsity component plus a 64-rank component decomposition, a compression scheme for which recent work shows impressive inference acceleration on GPUs, `HASSLE-free` reduces the test perplexity by $18\%$ for the WikiText-2 dataset and reduces the gap (compared to the dense model) of the average of eight popular zero-shot tasks by $28\%$ compared to existing methods. Our code is available at: https://github.com/mazumder-lab/HASSLE-free.

## 1. Introduction

Large Language Models (LLMs) have shown remarkable capabilities on numerous tasks in Natural Language Processing (NLP), ranging from language understanding to generation [1–4]. The huge success of LLMs comes with important challenges to deploy them due to their massive size and computational costs. For instance, Llama-3-405B [4] requires 780GB of storage in half precision (FP16) and hence multiple high-end GPUs are needed just for inference. *Model compression* has emerged as an important line of research to reduce the costs associated with deploying these foundation models. In particular, neural network pruning [5–7], where model weights are made to be sparse after training, has garnered significant attention. Different sparsity structures (Structured, Semi-Structured and Unstructured) obtained after neural network pruning result in different acceleration schemes. *Structured pruning* removes entire structures such as channels, filters, or attention heads [8–11] and readily results in acceleration as model weights dimensions are reduced. *Semi-Structured pruning*, also known as, N:M sparsity [12] requires that at most $N$ out of $M$ consecutive elements are non-zero elements. Modern NVIDIA GPUs provide support for 2:4 sparsity acceleration. *Unstructured pruning* removes individual weights [13, 14] from the model's weights and requires specialized hardware for acceleration. For instance, DeepSparse [15–17] provide CPU inference acceleration for unstructured sparsity.

Specializing to LLMs, one-shot pruning [18–21], where one does a single forward pass on a small amount of calibration data, and prunes the model without expensive fine-tuning/retraining, is of particular interest. This setup requires less hardware requirements. For instance, Meng et al. [18] show how to prune an OPT-30B [22] using a single consumer-level V100 GPU with 32GB of CUDA memory, whereas fine-tuning a pruned model (with a suboptimal sparsification strategy) using Adam [23] at half-precision requires more than 220GB of CUDA memory.

---

[*]Corresponding author <mmakni@mit.edu>

Second Conference on Parsimony and Learning (CPAL 2025).

An interesting new development in *model compression* is the sparse plus low-rank matrix decomposition problem which aims to approximate model's weights by a sparse component plus a low-rank component [24–32]. Specializing to LLMs, Zhang and Papyan [33] propose OATS that outperforms pruning methods for the same compression ratio (number of non-zero elements) on a wide range of LLM evaluation benchmarks (e.g. perplexity in Language generation).

OATS [33] is a matrix decomposition algorithm that draws inspiration from a pruning algorithm Wanda [20]. Wanda is related to another popular pruning method SparseGPT [19]. A recent work ALPS [18] shows that by directly optimizing a layer-wise reconstruction loss for pruning can result in better pruning-utility tradeoffs over prior approaches (e.g. Wanda, SparseGPT) especially for high-sparsity regimes. In this paper, motivated by ALPS [18], we provide an optimization framework to decompose pre-trained model weights into sparse plus low-rank components based on a layer-wise loss function. Our framework is modular and can incorporate different pruning and matrix-decomposition algorithms (developed independently in different contexts). We observe that our optimization-based framework results in models with better model utility-compression trade-offs, consistent with the findings in [18] for pruning. The advantage of our approach is particularly pronounced for higher compression regimes.

Concurrently, in a different and complementary line of work, [34] have open-sourced highly specialized CUDA kernels designed for N:M sparse [12] plus low-rank matrix decompositions that result in significant acceleration and memory reduction for the pre-training of LLMs. We note that our focus here is on improved algorithms for one-shot sparse plus low-rank matrix decompositions for foundation models with billions of parameters which is different from the work of [34] that focuses on accelerating the pre-training of LLMs. The designed CUDA kernels [34] can be exploited in our setting for faster acceleration and reduced memory footprint during inference.

Our focus here is also different from research related to contextual sparsity, which accelerates foundation models by cutting off entire specific attention heads or MLP parameters dynamically at inference time for a given input [35, 36]. The decomposition of LLMs' weights and contextual sparsity in LLMs are orthogonal research directions. They could be applied on top of each other in the sense that a pre-trained model can benefit from speedups if its layers are compressed in addition to speedups obtained thanks to contextual sparsity techniques which can be applied at inference time. Another difference and motivation for the static compression approach we consider here is that one can fine-tune the resulting model (with sparse plus low-rank decompositions) with LoRA (low rank adaptation [37]) by fixing the (semi-structured) sparse component and training the "smartly-initialized" low-rank components. This idea has been explored in the seminal works of [38, 39]. In our proposed setting, we can reap the benefits of a faster forward-pass thanks to available CUDA kernels for N:M sparsity plus low-rank structure and efficient backward-pass thanks to LoRA modules.

**Summary of approach.** Our framework is coined `HASSLE-free`: Hardware-Aware (Semi-Structured) Sparse plus Low-rank Efficient & approximation-free matrix decomposition for foundation models.

Hardware-aware refers to the fact that we mostly focus on a N:M sparse [12] plus low-rank decomposition, for which acceleration on GPUs is possible, although `HASSLE-free` supports any type of sparsity pattern (unstructured, semi-structured, structured) in the sparsity constraint. Approximation-free refers to the fact that we consider minimizing the local layer-wise reconstruction error introduced in Equation (1). We show that prior work considers an approximation of this objective.

We formulate the compression/decomposition task as a clean optimization problem: we minimize a local layer-wise reconstruction objective where the weights are given by the sum of a sparse and a low-rank component. We propose an efficient alternating minimization approach that scales to models with billions of parameters relying on two key components: one involving sparse minimization (weight sparsity) and the other involving a low-rank optimization. We discuss how prior algorithms can be interpreted as approximately solving these subproblems, each with different approximation schemes.

We note that `HASSLE-free` differs from prior one-shot (sparse) pruning methods such as [7, 18, 19, 40, 41] as we seek a sparse plus low-rank decomposition of weights. Additionally, it differs from prior one-shot sparse plus low-rank matrix decomposition methods [33] as we directly minimize the local layer-wise reconstruction objective introduced in Equation (1).

Our main **contributions** can be summarized as follows.

- We introduce `HASSLE-free` a unified one-shot LLM compression framework that scales to models with billions of parameters where we directly minimize the local layer-wise reconstruction error subject to a sparse plus low-rank matrix decomposition of the pre-trained dense weights.
- `HASSLE-free` uses an alternating minimization approach that optimizes over a sparse and a low-Rank component. `HASSLE-free` can use any pruning method as a plug-in for the subproblem pertaining to the sparse component. Additionally, it uses gradient-descent type methods to optimize the subproblem pertaining to the low rank component.
- We discuss how special cases of our framework relying on specific approximations of the objective retrieve popular methods such as OATS, Wanda and MP — [13, 20, 33, 42]. This provides valuable insights into the underlying connections across different methods.
- `HASSLE-free` improves upon state-of-the-art methods for one-shot sparse plus low-rank matrix decomposition. For the Llama3-8B model with a 2:4 sparsity component plus a 64-rank component decomposition, `HASSLE-free` reduces the test perplexity by $18\%$ for the WikiText-2 dataset and reduces the gap (compared to the dense model) of the average of eight popular zero-shot tasks by $28\%$ compared to existing methods.

## 2. Related Work

**Network pruning.** Network pruning is a popular technique for reducing the complexity of deep neural networks by removing redundant weights [5, 13]. Pruning methods can be classified on the basis of the structure of the resulting sparse network. In terms of structure, pruning can be categorized into unstructured pruning, semi-structured pruning, and structured pruning. Unstructured pruning offers better flexibility and higher sparsity levels, but requires specialized hardware for acceleration, while structured pruning is more hardware-friendly but may suffer from larger performance degradation. Semi-structured sparsity combines the benefits of unstructured sparsity in terms of retaining the model's performance thanks to its flexibility and the benefits of structured sparsity in terms of efficiency. For example, NVIDIA has recently introduced sparse tensor cores [34] to their hardware that accelerate Gemm with N:M sparsity on modern NVIDIA GPUs. In this paper, we mostly consider N:M sparsity [12] for the sparsity constraint, although `HASSLE-free` supports other sparsity structures.

Recently, algorithms —-inspired by large-scale mathematical optimization tools—have been proposed to prune a large pre-trained network under sparsity constraints. For example, CHITA [7], FALCON [40] consider pruning using a Fisher loss function under unstructured sparsity and/or FLOP constraints; ALPS [18] consider pruning using a layerwise reconstruction loss function under unstructured/semi-structured sparsity; OSSCAR [43] study structured sparsity using a layerwise reconstruction loss. Recently, [41] present SNOWS studying pruning for vision models using a specialized loss function that takes into account higher-order feature embeddings. We refer the reader to related work discussed in the aforementioned earlier papers for other nice algorithmic work on pruning.

**Sparse plus Low-Rank Matrix Decomposition.** Decomposing a weight matrix into a low-rank matrix plus a sparse matrix—closely related to the "robust PCA" problem—is a well-studied problem from both theoretical and algorithmic perspectives in statistics, signal processing, optimization communities [24–28]. More recently, these approaches have been explored in [29, 30] and in deep learning by [31]. They have been extended to LLMs by [32] in the context of improving the utility of fine-tuning LLMs and by [33] for sparse plus low-rank model compression.

**One-shot matrix decompositions in LLMs.** Matrix decomposition in the context of LLMs has gathered a lot of attention recently. [38, 39] decompose models' weights into a quantized weight plus a low-rank component. [39] study an alternating-minimization approach in a data-free fashion (without using a calibration dataset). [38] consider both a data-free and a data-aware decomposition for the quantized plus low-rank decomposition problem. Their data-aware decomposition relies on an approximation of the Fisher importance matrix. The OATS approach [33] considers a sparse plus low-rank decomposition of model's weights—they take inspiration from the pruning algorithm Wanda [20] to incorporate outlier information (from a calibration dataset) in their decomposition. Our paper generalizes OATS [33] and computes a decomposition incorporating more information from the calibration dataset.

## 3. Problem Formulation

We first introduce some notation that we will use throughout the paper.

**Notation.** For a matrix $\mathbf{Z} \in \mathbb{R}^{m \times n}$, $\mathrm{rk}(\mathbf{Z})$ denotes the rank of $\mathbf{Z}$. For a given rank $r \in \mathbb{N}$, denote $C_r(\mathbf{Z}) = \mathbf{U}_r \boldsymbol{\Sigma}_r \mathbf{V}_r^T$, corresponding to the matrices formed by retaining only the top-$r$ singular vectors and singular values from the full SVD of $\mathbf{Z}$. The Eckart and Young [44] theorem shows that $C_r(\mathbf{Z}) = \arg\min_{\mathbf{M}:\mathrm{rk}(\mathbf{M})\leq r} \|\mathbf{Z} - \mathbf{M}\|_F$ where, $\|\cdot\|_F$ denotes Frobenius norm.

For a square matrix $\mathbf{Z} \in \mathbb{R}^{n \times n}$, $\mathrm{Tr}(\mathbf{Z}) = \sum_{i \in [n]} \mathbf{Z}_{ii}$ denotes the trace of $\mathbf{Z}$, $\mathrm{diag}(\mathbf{Z})$ denotes the diagonal matrix $\mathbf{D} \in \mathbb{R}^{n \times n}$ such that $\mathbf{D}_{ii} = \mathbf{Z}_{ii}$ for any $i \in [n]$, and $\mathbf{D}_{ij} = 0$ for any $i \neq j$, $i, j \in [n]$. We also let $\mathbf{1}_n$ (and $\mathbf{0}_n$) denote the vector of all ones (and all zeros) of length $n \in \mathbb{N}$.

**Layer-wise Reconstruction Error.** Building on the one-shot pruning framework of Frantar and Alistarh [19], we decompose the foundation model into layer-wise sub-problems, to be solved sequentially layer by layer, where one aims to minimize the $\ell_2$ error (ie, Frobenius norm error) between the outputs of a dense layer and that of its compressed counterpart.

For each layer-wise sub-problem, let $\widehat{\mathbf{W}} \in \mathbb{R}^{N_{\mathrm{in}} \times N_{\mathrm{out}}}$ denote the pre-trained (dense) weight matrix of layer $\ell$, where $N_{\mathrm{in}}$ and $N_{\mathrm{out}}$ denote the input and output dimension of the layer, respectively. Given a set of $N$ calibration samples, the input activation matrix can be represented as $\mathbf{X} \in \mathbb{R}^{NL \times N_{\mathrm{in}}}$, where $L$ is the sequence length of an LLM. It corresponds to the output of the previous layer $(\ell - 1)$ compressed layers (sparse plus low-rank), computed using the $N$ calibration samples. We seek to find a sum of a sparse weight matrix $\mathbf{W_S}$ and a low-rank weight matrix $\mathbf{M}$ that minimizes the reconstruction error between the original and new layer outputs, while satisfying a target sparsity constraint and a low-rank constraint. The optimization problem is given by

$$\min_{\mathbf{W_S}, \mathbf{M}} \left\| \mathbf{X}\widehat{\mathbf{W}} - \mathbf{X}(\mathbf{W_S} + \mathbf{M}) \right\|_F^2 \quad \text{s.t.} \quad \mathbf{W_S} \in \mathcal{C}_S, \quad \mathrm{rk}(\mathbf{M}) \leq r. \tag{1}$$

where $\mathbf{W_S}, \mathbf{M} \in \mathbb{R}^{N_{\mathrm{in}} \times N_{\mathrm{out}}}$, $\mathcal{C}_S$ denotes the the sparsity-pattern constraint set.

## 4. Algorithm Design

Optimizing Problem (1) is challenging as the constraints are non convex. While this optimization formulation relates to the Robust-PCA literature [24, 25, 45], the size of parameters in $\mathbf{W_S}$ and $\mathbf{M}$ can reach over 100 million in the LLM setting. For instance, the size of a down projection in a FFN of a Llama3-405b [4] has more than 800 million parameters. Due to the limitations of prior approaches (at this scale), we need to design computationally efficient algorithms to address the layer-wise reconstruction matrix decomposition problem (1).

We propose to (approximately) optimize problem (1) using an alternating minimization approach [24, 28]. At each iteration, we consider two sub-problems. In particular, at iteration $t$, we consider sub-problem (**P1**), which pertains to the sparse component of the matrix decomposition:

$$\mathbf{W_S}^{(t+1)} \in \arg\min_{\mathbf{W_S}} \left\| \mathbf{X}\widehat{\mathbf{W}} - \mathbf{X}\left(\mathbf{W_S} + \mathbf{M}^{(t)}\right) \right\|_F^2 \quad \text{s.t.} \quad \mathbf{W_S} \in \mathcal{C}_S \tag{2}$$

$$= \arg\min_{\mathbf{W_S}} \left\| \mathbf{X}\tilde{\mathbf{W}}^{(t)} - \mathbf{X}\mathbf{W_S} \right\|_F^2 \quad \text{s.t.} \quad \mathbf{W_S} \in \mathcal{C}_S. \qquad (\tilde{\mathbf{W}}^{(t)} := \widehat{\mathbf{W}} - \mathbf{M}^{(t)})$$

The second sub-problem we consider, at iteration $t$, pertains to the low-rank component of the matrix decomposition problem, is (**P2**):

$$\mathbf{M}^{(t+1)} \in \arg\min_{\mathbf{M}} \ \left\| \mathbf{X}\widehat{\mathbf{W}} - \mathbf{X}\left(\mathbf{W_S}^{(t+1)} + \mathbf{M}\right)\right\|_F^2 \quad \text{s.t.} \quad \text{rk}\left(\mathbf{M}\right) \le r \tag{3}$$

$$= \arg\min_{\mathbf{M}} \ \left\| \mathbf{X}\bar{\mathbf{W}}^{(t+1)} - \mathbf{X}\mathbf{M}\right\|_F^2 \quad \text{s.t.} \quad \text{rk}\left(\mathbf{M}\right) \le r. \qquad (\bar{\mathbf{W}}^{(t+1)} := \widehat{\mathbf{W}} - \mathbf{W_S}^{(t+1)})$$

For notation simplicity, we remove the dependence on the iteration $t$ and study (2) rewritten as follows:

$$\mathbf{W_S}^{\star} \in \arg\min_{\mathbf{W_S}} \ \left\| \mathbf{X}\tilde{\mathbf{W}} - \mathbf{X}\mathbf{W_S}\right\|_F^2 \quad \text{s.t.} \quad \mathbf{W_S} \in \mathcal{C}_S \tag{4}$$

$$= \arg\min_{\mathbf{W_S}} \ \text{Tr}\left((\tilde{\mathbf{W}} - \mathbf{W_S})^{\top}\mathbf{H}(\tilde{\mathbf{W}} - \mathbf{W_S})\right) \quad \text{s.t.} \quad \mathbf{W_S} \in \mathcal{C}_S. \qquad (\mathbf{H} = \mathbf{X}^{\top}\mathbf{X})$$

Similarly, we study (3) rewritten as follows:

$$\mathbf{M}^{\star} \in \arg\min_{\mathbf{M}} \ \left\| \mathbf{X}\bar{\mathbf{W}} - \mathbf{X}\mathbf{M}\right\|_F^2 \quad \text{s.t.} \quad \text{rk}\left(\mathbf{M}\right) \le r \tag{5}$$

$$= \arg\min_{\mathbf{M}} \ \text{Tr}\left((\bar{\mathbf{W}} - \mathbf{M})^{\top}\mathbf{H}(\bar{\mathbf{W}} - \mathbf{M})\right) \quad \text{s.t.} \quad \text{rk}\left(\mathbf{M}\right) \le r.$$

We proceed to discuss algorithms that solve different variations of (**P1**) and (**P2**), and establish connections across existing methods in *model compression*.

## 4.1. Minimizing sub-problem (P1)

For (4), one can consider different variants for $\mathbf{H}$, the Hessian of the local layer-wise reconstruction error.

### 4.1.1. Data-Free version: $\mathbf{X} = \mathbf{I}_{N_{\text{in}} \times N_{\text{in}}} \implies \mathbf{H} = \mathbf{I}_{N_{\text{in}} \times N_{\text{in}}}$

A data-free pruning method (without a calibration dataset) considers $\mathbf{X}$ to be an identity matrix in (4). When $\mathbf{H}$ is identity matrix, equation (4) can be solved to optimality and an optimal solution is obtained with Magnitude Pruning (MP, [13, 42]) using a simple hard thresholding operator on the dense weight $\tilde{\mathbf{W}}$ – keeping the largest values and setting the remaining values to zero. Note that MP can be applied to unstructured [13], semi-structured N:M sparsity [12], and structured pruning [18]. This accommodates most sparsity sets $\mathcal{C}_S$ in the pruning literature.

### 4.1.2. Diagonal-approximation: $\mathbf{H} = \text{diag}(\mathbf{X}^{\top}\mathbf{X})$

In Problem (4) we can approximate the Hessian of the local layer-wise reconstruction error by its diagonal. An optimal solution in this case, can be obtained by hard thresholding $\mathbf{D}\tilde{\mathbf{W}}$, where $\mathbf{D} = \sqrt{\text{diag}(\mathbf{X}^{\top}\mathbf{X})}$. Note that this approximation results in the state-of-the-art pruning algorithm Wanda [20]. In fact, the importance metric, $S_{ij}$ in Wanda for each entry $\tilde{\mathbf{W}}_{ij}$ is given by:

$$S_{ij} = \left|\tilde{\mathbf{W}}_{ij}\right| \cdot \|\mathbf{X}_i\|_2 = \left|\mathbf{D}\tilde{\mathbf{W}}\right|_{ij} \qquad (\mathbf{D} = \sqrt{\text{diag}(\mathbf{X}^{\top}\mathbf{X})})$$

where, $\mathbf{X}_i$ [2] denotes the $i^{th}$ column of the input activation matrix $\mathbf{X}$. [20] show impressive results with this approximation for unstructured and semi-structured sparsity. OATS [33] decomposes model weights into sparse plus low-rank using alternating minimization; their sparse update uses this Wanda approximation (diagonal of the local layer-wise objective's Hessian).

### 4.1.3. Full Hessian: $\mathbf{H} = \mathbf{X}^{\top}\mathbf{X} + \lambda \mathbf{I}$

This approach aims to minimize (4). [19] consider the full Hessian of the local layer-wise reconstruction objective (**P1**) at the scale of pruning LLMs—they use approximations to simplify the algorithm (as opposed to approximating the optimization formulation). Meng et al. [18] propose

---

[2]The difference in this formula–indexing of columns of $\mathbf{X}$– with respect to the one introduced the Wanda paper stems from the fact that Wanda considers the output of the layer to be $\mathbf{X}\mathbf{W}^{\top}$ whereas we consider $\mathbf{X}\mathbf{W}$.

advanced algorithms to obtain good solutions to this optimization program using the operator splitting technique ADMM [46]—they find impressive results for unstructured sparsity and N:M sparsity. Meng et al. [43] consider the full Hessian formulation for structured sparsity and use combinatorial optimization techniques for optimization.

Our framework can use any pruning algorithm developed for minimizing (**P1**). Since we aim to directly optimize formulation (1), we select methods that use the entire Hessian—we observe that these approaches give better utility for high compression ratios. SparseGPT [19] and ALPS [18] are high-quality pruning methods that consider the entire Hessian. For our numerical results, we present results using SparseGPT (default) and/or ALPS for problem (**P1**).

## 4.2. Minimizing sub-problem (P2)

For problem (**P2**), we discuss algorithms and related work for different choices of $\mathbf{H}$.

### 4.2.1. Data-Free version: $\mathbf{X} = \mathbf{I}_{N_{\text{in}} \times N_{\text{in}}}$

Similar to the pruning literature, we consider a data-free version for the rank constrained problem. Here a closed-form solution of the minimizer is given by the Truncated-SVD $C_r(\bar{\mathbf{W}})$ corresponding to the best rank-$r$ approximation of $\bar{\mathbf{W}}$. Li et al. [39] use SVD on the full matrix during their low-rank minimization step for quantization plus low-rank matrix decomposition. Guo et al. [38] use a randomized SVD [47] (in context of quantization plus low-rank decomposition) instead of the full SVD resulting in runtime improvements.

### 4.2.2. Diagonal-approximation: $\mathbf{H} = \text{diag}(\mathbf{X}^\top \mathbf{X})$

The diagonal approximation of $\mathbf{H}$ appears in the pruning literature, see eg Wanda [20]. We analyze problem (**P2**) with this approximation. Similar to 4.1.2, we introduce $\mathbf{D} = \sqrt{\text{diag}(\mathbf{X}^\top \mathbf{X})}$ in (5), use the fact that $\mathbf{D}$ is symmetric, and have:

$$\mathbf{M}^\star \in \arg\min_{\mathbf{M}} \ \text{Tr}\left((\bar{\mathbf{W}} - \mathbf{M})^\top \mathbf{D}^2 (\bar{\mathbf{W}} - \mathbf{M})\right) \quad \text{s.t.} \quad \text{rk}\,(\mathbf{M}) \leq r$$
$$= \arg\min_{\mathbf{M}} \ \left\|\mathbf{D}\bar{\mathbf{W}} - \mathbf{D}\mathbf{M}\right\|_F^2 \quad \text{s.t.} \quad \text{rk}\,(\mathbf{M}) \leq r.$$

**Assumption 1** *The input activations matrix $\mathbf{X}$ satisfies $\text{diag}\left(\mathbf{X}^\top \mathbf{X}\right)$ is full-rank. Equivalently, no column of $\mathbf{X}$ is identically $\mathbf{0}_{N \cdot L}$.*

**Theorem 4.1** *If Assumption 1 holds, then the closed-form minimizer of (5) is given by*

$$\mathbf{M}^\star = \mathbf{D}^{-1} C_r(\mathbf{D}\bar{\mathbf{W}}).$$

The proof of Theorem 4.1 is obtained by introducing the auxiliary variable $\tilde{\mathbf{M}} = \mathbf{D}\mathbf{M}$ and noting that $\text{rk}(\tilde{\mathbf{M}}) = \text{rk}(\mathbf{M})$, when **Assumption 1** holds.

Interestingly, OATS [33] uses the same operation in their low rank update as a part of the alternating minimization approach (for sparse plus low rank matrix decomposition).

**Corollary 4.2** *OATS [33] exactly minimizes (1) with a diagonal approximation of the Hessian of the local layer-wise reconstruction error, since they minimize (**P1**) and (**P2**) with the same diagonal approximation $\mathbf{H} = \text{diag}(\mathbf{X}^\top \mathbf{X})$.*

### 4.2.3. Full Hessian: $\mathbf{H} = \mathbf{X}^\top \mathbf{X} + \lambda \mathbf{I}$

Motivated by the use of a full Hessian for pruning (cf Sec 4.1.3), we also consider Problem (5) by using the full Hessian. We reparametrize the low-rank matrix $\mathbf{M} \in \mathbb{R}^{N_{\text{in}} \times N_{\text{out}}}$ by $\mathbf{U}\mathbf{V}^\top$, with $\mathbf{U} \in \mathbb{R}^{N_{\text{in}} \times r}, \mathbf{V} \in \mathbb{R}^{N_{\text{out}} \times r}$. We use first-order optimization methods to (approximately) minimize the layer-wise reconstruction objective (wrt $\mathbf{U}, \mathbf{V}$), given by:

$$\mathbf{M}^\star = \mathbf{U}^\star \mathbf{V}^{\star\top}, \quad \mathbf{U}^\star, \mathbf{V}^\star \in \arg\min_{\mathbf{U}, \mathbf{V}} \ \text{Tr}\left(\left(\bar{\mathbf{W}} - \mathbf{U}\mathbf{V}^\top\right)^\top \mathbf{H} \left(\bar{\mathbf{W}} - \mathbf{U}\mathbf{V}^\top\right)\right). \tag{6}$$

**Diagonal Scaling for Numerical Stability.** Our initial experiments for problem (6), using gradient descent type methods on $\mathbf{U}$ and $\mathbf{V}$ showed that the optimization problem can be ill-conditioned in some transformer layers. This can lead to numerical instability in the optimization procedure. To address this, we follow a similar rescaling approach proposed by Meng et al. [18]. Define (similar to 4.2.2) the matrix $\mathbf{D} = \sqrt{\mathrm{diag}(\mathbf{X}^\top \mathbf{X})}$ and reformulate the optimization equation (6) as follows (when **Assumption**1 holds).

$$\mathbf{M}^\star = \mathbf{D}^{-1}\mathbf{U}^\star \mathbf{V}^{\star\top}, \quad \mathbf{U}^\star, \mathbf{V}^\star \in \arg\min_{\mathbf{U},\mathbf{V}} \ \mathrm{Tr}\left( \left( \mathbf{D}\bar{\mathbf{W}} - \mathbf{U}\mathbf{V}^\top \right)^\top \mathbf{D}^{-1}\mathbf{H}\mathbf{D}^{-1} \left( \mathbf{D}\bar{\mathbf{W}} - \mathbf{U}\mathbf{V}^\top \right) \right). \tag{7}$$

We note that the minimization problems in (7) and (6) are equivalent. This scaling, which sets the diagonal of the new Hessian to $\mathbf{1}_{N_{\mathrm{in}}}$, only modifies the steps of gradient descent and leads to faster convergence in practice. See Figure 1 showing the usefulness of our proposed diagonal scaling.

## 4.3. Our Proposed Approach

We consider program (1) with the full Hessian. Our results show that using the entire Hessian outperforms OATS [33], which considers (1) with the diagonal approximation of the Hessian approach, on a wide range of LLM benchmarks and compression ratios.

In our experiments, we show results with the SparseGPT [19] (default) or ALPS [18] algorithm to minimize (**P1**) and the Adam algorithm [23] to minimize (**P2**) reparameterized and rescaled as in (7). For (**P1**), we let `Ours w/ SparseGPT` and `Ours w/ ALPS` denote the algorithms that use `HASSLE-free` with SparseGPT and ALPS (respectively).

**Optimizations for Efficiency.** Note that for a given layer $\ell$, the Hessian of the local layer-wise reconstruction problem $\mathbf{X}^\top\mathbf{X}$ in (4) as well as the rescaled version $\mathbf{D}^{-1}\mathbf{X}^\top\mathbf{X}\mathbf{D}^{-1}$ in (5) are invariant throughout iterations. This is very important as pruning algorithms that use the entire Hessian information [18, 19] need the Hessian inverse in their algorithm update. This inversion and associated costs of Hessian construction are done only once (for each layer) and then *amortized* throughout iterations. In Algorithm 1, we use $\mathbf{U}^{(t-1)}$ and $\mathbf{V}^{(t-1)}$ as initializations for the optimizer, as they are close to the minimizers of (**P2**) at iteration $t$. This accelerates the convergence in practice.

**Computational Complexity of `HASSLE-free w/ SparseGPT` & Runtimes.**

- Hessian construction in $O(NLN_{\mathrm{in}})$: This is obtained by using the identity $\mathbf{X}^\top\mathbf{X} = \sum_{i=1}^{NL} x_i x_i^\top$ following [19].
- Hessian inversion in $O(N_{\mathrm{in}}^3)$.
- Sparse minimization (w/ `SparseGPT`[3]) in $O(T_{\mathrm{AM}}(N_{\mathrm{in}}^3 + N_{\mathrm{in}}^2 N_{\mathrm{out}}))$: The associated pruning itself is applied during each of the alternating minimization steps, hence its cost is multiplied by $T_{\mathrm{AM}}$.
- Low-Rank minimization in $O(T_{\mathrm{AM}}T_{\mathrm{LR}}N_{\mathrm{in}}^2 N_{\mathrm{out}})$: The first-order optimization of $\mathbf{U}, \mathbf{V}$ is performed in $O(N_{\mathrm{in}}^2 N_{\mathrm{out}})$ for $T_{\mathrm{AM}}T_{\mathrm{LR}}$ times overall.

| Model | Algorithm | Runtime |
|-------|-----------|---------|
| | OATS-2:4+64LR | 9.27 |
| Llama3-8B | Ours-2:4+64LR w/ SparseGPT | 20.49 |
| | Ours-2:4+64LR w/ ALPS | 20.13[4] |
| | OATS-2:4+64LR | 0.45 |
| Llama3.2-1B | Ours-2:4+64LR w/ SparseGPT | 2.16 |
| | Ours-2:4+64LR w/ ALPS | 6.77 |
| | OATS-2:4+64LR | 2.81 |
| Llama3.2-3B | Ours-2:4+64LR w/ SparseGPT | 6.19 |
| | Ours-2:4+64LR w/ ALPS | 17.06 |

Table 1: Runtime (hours) Analysis for one-shot 2:4 sparse plus a 64-rank matrix decomposition of `HASSLE-free`. All unmarked experiments were run on a single L40 GPU.

In the context of LLMs, where $N_{\mathrm{in}}, N_{\mathrm{out}}$ are a constant multiple of $h$, the LLM hidden dimension, the complexity of `HASSLE-free w/ SparseGPT` is given by $O(NLh + T_{\mathrm{AM}}T_{\mathrm{LR}}h^3)$.

---

[3]The computational cost of ALPS is more involved – we report only its runtime in Table 1

[4]Using a single A100 80GB GPU. We use L40 48GB GPUs for all other experiments in Table 1.

---
**Algorithm 1** `Low-Rank-GD`
---

**Input** `Optimizer` (optimization algorithm, e.g. Adam), $\mathbf{H}$ (Hessian), $\mathbf{W}$ (Weights), $\mathbf{U}_{\text{init}}$, $\mathbf{V}_{\text{init}}$ (warm-up initialization for the joint minimization of $\mathbf{U}, \mathbf{V}$), $T_{\text{LR}}$ (# iterations), $\eta$ (learning rate).

$\text{Obj}(\mathbf{U}, \mathbf{V}) \leftarrow \text{Tr}\left( (\mathbf{W} - \mathbf{U}\mathbf{V}^\top)^\top \mathbf{H} (\mathbf{W} - \mathbf{U}\mathbf{V}^\top) \right)$

$\mathbf{U}^\star, \mathbf{V}^\star \leftarrow \texttt{Optimizer}_{\mathbf{U},\mathbf{V}} \left( \text{Obj}, \mathbf{U}_{\text{init}}, \mathbf{V}_{\text{init}}, T_{\text{LR}}, \eta \right)$

**Output** $\mathbf{U}^\star, \mathbf{V}^\star$.

---
**Algorithm 2** `HASSLE-free`
---

**Input** for a given layer $\ell$: $\mathbf{H} = (\mathbf{X}^\top\mathbf{X} + \lambda\mathbf{I})$ (Hessian of (1), plus a regularization term for numerical stability), $\widehat{\mathbf{W}}$ (dense pre-trained weights), $T_{\text{AM}}$ (# iterations of alternating-minimization), $T_{\text{LR}}$ (# iterations of `Low-Rank-GD`), $\eta$ (learning rate for $\mathbf{U}, \mathbf{V}$), $\mathcal{C}_S$ (sparsity pattern), $r$ (rank of low-rank components), `Prune` (any pruning algorithm, e.g. SparseGPT/ALPS), `Optimizer` (any first-order algorithm, e.g. Adam), **is_scaled** (bool to apply scaling 4.2.3).

$\mathbf{D} \leftarrow \sqrt{\text{diag}(\mathbf{H})}$     `// Diagonal of the Hessian.`
$\mathbf{H}^{-1} \leftarrow \texttt{inv}(\mathbf{H})$     `// Inverse the Hessian.`
$\mathbf{W_S} \leftarrow \mathbf{0}_{N_{\text{in}} \times N_{\text{out}}}$
$\mathbf{U} \leftarrow \mathbf{0}_{N_{\text{in}} \times r}$
$\mathbf{V} \leftarrow \mathcal{N}_{N_{\text{out}} \times r}$     `// element-wise independent gaussian initialization.`
**for** $t = 1 \dots T_{\text{AM}}$ **do**
    $\mathbf{W_S} \leftarrow \texttt{Prune}\left( \mathbf{H}^{-1}, \widehat{\mathbf{W}} - \mathbf{U}\mathbf{V}^\top, \mathcal{C}_S \right)$
    `// `$\mathbf{W_S} \approx \widehat{\mathbf{W}} - \mathbf{U}\mathbf{V}^\top$`, satisfies `$\mathcal{C}_S$` sparsity pattern & minimizes (P1).`
    $\eta_t \leftarrow \text{get\_lr}(t, \eta)$     `// In practice, `$\eta_t = \eta/(t + 10)$`.`
    **if is_scaled then**
       $\mathbf{U}, \mathbf{V} \leftarrow \texttt{Low-Rank-GD}\left( \texttt{Optimizer}, \mathbf{D}^{-1}\mathbf{H}\mathbf{D}^{-1}, \mathbf{D}\left( \widehat{\mathbf{W}} - \mathbf{W_S} \right), \mathbf{D}\mathbf{U}, \mathbf{V}, T_{\text{LR}}, \eta_t \right)$
       $\mathbf{U} \leftarrow \mathbf{D}^{-1}\mathbf{U}$     `// Rescale `$\mathbf{U}$` back.`
    **else**
       $\mathbf{U}, \mathbf{V} \leftarrow \texttt{Low-Rank-GD}\left( \texttt{Optimizer}, \mathbf{H}, \widehat{\mathbf{W}} - \mathbf{W_S}, \mathbf{U}, \mathbf{V}, T_{\text{LR}}, \eta_t \right)$
    `// `$\mathbf{U}\mathbf{V}^\top \approx \widehat{\mathbf{W}} - \mathbf{W_S}$`, has rank at most `$r$` & minimizes (P2).`
$\mathbf{M} \leftarrow \mathbf{U}\mathbf{V}^\top$
**Output** for a given layer $\ell$: $\mathbf{W_S}, \mathbf{M}$.

---

# 5. Experimental Results

## 5.1. Experiment Setup

**Models and datasets** We evaluate our proposed method `HASSLE-free` on two families of large language models: Llama-3 and Llama-3.2 [4] with sizes ranging from 1 to 8 billion parameters. To construct the Hessian $\mathbf{X}^\top\mathbf{X}$, we follow [19]: we use 128 segments of 2048 sequence length each, randomly sampled from the first shard of the C4 training dataset [48]. To ensure consistency, we use the same calibration data for all pruning algorithms we benchmark. We also consider one-shot compression results without retraining. We assess the performance using perplexity and zero-shot evaluation benchmarks, with perplexity calculated according to the procedure described by HuggingFace [49], using full stride. For perplexity evaluations, we use the test sets of raw-WikiText2 [50], PTB [51], and a subset of the C4 validation data, which are popular benchmarks in the LLM pruning literature [18, 19, 43]. Additionally, we evaluate the following zero-shot tasks using LM Harness by Gao et al. [52]: PIQA [53], ARC-Easy (ARC-E) & ARC-Challenge (ARC-C) [54], Hellaswag (HS) [55], Winogrande (WG) [56], RTE [57], OpenbookQA (OQA) [58] and BoolQ [59]. The average of the eight zero-shot tasks is also reported.

## 5.2. Results

To benchmark the performance of our matrix decomposition algorithm, `HASSLE-free` uses the same number of alternating-minimization steps as OATS [33] which is $80$. We report results for the scaled version of `HASSLE-free`, with the same learning rate $\eta = 1e^{-2}$ for all layers and considered models. We consider the following two settings.

**N:M Sparsity + Fixed Rank:** We impose the sparsity pattern $\mathcal{C}_S$ to be $N : M$ sparsity and we fix the target rank $r = 64$ of the low-rank component for all layers. We benchmark our method with OATS [33]. The results are reported in Table 3.

**N:M Sparsity + Fixed Compression Ratio:** This is similar to the setting described by [33] for N:M sparsity evaluations. Each layer, with dense weight matrix $\widehat{\mathbf{W}}$, is compressed to a prefixed compression ratio $\rho$ (e.g. $50\%$) so that $\widehat{\mathbf{W}} \approx \mathbf{W}_{N:M} + \mathbf{M}$, and the target rank is given by $r = \left\lfloor (1 - \rho - \frac{N}{M}) \cdot (N_{\text{out}} \cdot N_{\text{in}}) / (N_{\text{out}} + N_{\text{in}}) \right\rfloor$.

Note that the effective number of parameters stored is therefore

$$\#\text{params } \mathbf{W}_{N:M} + \#\text{params } \mathbf{U} + \#\text{params } \mathbf{V} = \frac{N}{M} \cdot (N_{\text{out}} \cdot N_{\text{in}}) + r N_{\text{in}} + r N_{\text{out}} \leq (1 - \rho) \cdot \#\text{params } \widehat{\mathbf{W}},$$

hence the comparison to other pruning methods matched at the same compression ratio $\rho$. The results are reported for the Llama3-8B model in Table 2 for `HASSLE-free`, `OATS`, and different N:M pruning algorithms (SparseGPT [19], Wanda [20], DSNoT [21]) compressed at $\rho = 50\%$. The results are expanded for `HASSLE-free` and OATS in Appendix A.

Table 2: Performance analysis for one-shot N:M sparse plus a low-rank matrix decomposition of the Llama3-8b model. The compression ratio is fixed to be $\rho = 0.5$. For Perplexity, ($\downarrow$) lower values are preferred. For zero-shot tasks, ($\uparrow$) higher values are preferred. Bolded values correspond to a comparison between sparse plus low-rank decomposition algorithms. Underlined values correspond to the overall best comopression scheme given a compression ratio $\rho = 50\%$.

| Algorithm | Perplexity ($\downarrow$) | | | Zero-shot ($\uparrow$) | | |
|---|---|---|---|---|---|---|
| | C4 | WT2 | PTB | PIQA | ARC-E | ARC-C |
| SparseGPT-4:8 | 14.94 | 12.40 | 17.90 | 73.20 | 68.54 | 34.86 |
| Wanda-4:8 | 18.88 | 14.52 | 24.26 | 71.52 | 64.91 | 34.03 |
| DSNoT-4:8 | 18.89 | 14.76 | 23.90 | 71.49 | 65.65 | 33.57 |
| SparseGPT-2:4 | 18.89 | 16.35 | 25.08 | 70.54 | 63.09 | 31.84 |
| Wanda-2:4 | 30.81 | 24.36 | 44.89 | 67.56 | 56.20 | 26.11 |
| DSNoT-2:4 | 28.78 | 23.09 | 40.95 | 67.70 | 56.46 | 25.68 |
| OATS-2:8+LR | 21.03 | 14.54 | 24.15 | 73.67 | 59.68 | 37.12 |
| Ours-2:8+LR w/ SparseGPT | 20.05 | 15.03 | 22.01 | 74.05 | 60.52 | 36.18 |
| Ours-2:8+LR w/ ALPS | **17.89** | **13.07** | **19.11** | **74.54** | **65.53** | **39.08** |
| OATS-3:8+LR | 16.87 | 11.43 | 18.53 | 75.24 | 65.91 | 39.85 |
| Ours-3:8+LR w/ SparseGPT | 16.16 | 11.36 | 16.71 | 75.79 | 67.55 | 41.04 |
| Ours-3:8+LR w/ ALPS | **14.85** | **10.20** | **15.42** | **77.15** | **69.40** | **43.64** |
| dense | 9.44 | 6.14 | 11.18 | 80.79 | 77.69 | 53.33 |

## 5.3. Reconstruction error on a single Transformer block

To show the performance of OATS and `HASSLE-free` on the layer-wise reconstruction objective (1), we compute the error produced with the two algorithms (both after $80$ iterations–default value used in OATS [33]). This is given by $\|\mathbf{X}\widehat{\mathbf{W}} - \mathbf{X}(\mathbf{W_S} + \mathbf{M})\|_F^2$, when applied to the model Llama-3-8B [4], we let $\mathcal{C}_S$ correspond to $2 : 4$ sparsity and use fixed rank $r = 64$. Results of the local layer-wise error are reported in Figure 1 for OATS, `HASSLE-free` scaled and `HASSLE-free` unscaled.

Figure 1: Local layer-wise reconstruction error $\downarrow$ (lower values are preferred) analysis of the decomposition of the layers of the **first** transformer block in Llama-3-8B into a 2:4 sparse component plus a 64-rank low-rank component. All methods use the same number of alternating minimization steps 80. We show results for `HASSLE-free` w/ SparseGPT for minimizing (**P1**).

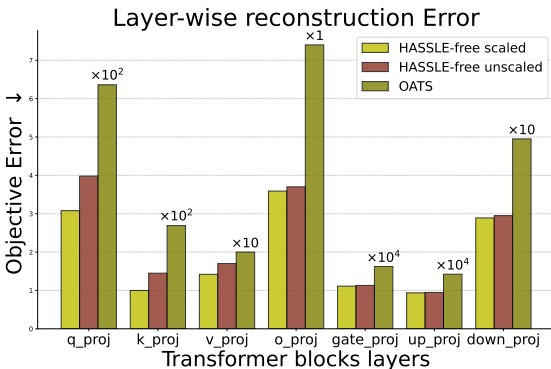

## 6. Conclusion

We present `HASSLE-free`, a unified framework for one-shot sparse plus low-rank matrix decomposition for foundation models. `HASSLE-free` considers a local layer-wise reconstruction objective and

| Model | Algorithm | Perplexity (↓) | | | Zero-shot (↑) | | | | | | | | |
|---|---|---|---|---|---|---|---|---|---|---|---|---|---|
| | | C4 | WT2 | PTB | PIQA | HS | ARC-E | ARC-C | WG | RTE | OQA | BoolQ | Avg |
| Llama3-8B | OATS-2:8+64LR | 368.24 | 416.14 | 565.46 | 52.29 | 28.03 | 27.53 | **22.70** | 49.17 | **52.71** | 26.40 | 42.08 | 37.61 |
| | Ours-2:8+64LR w/ SparseGPT | 90.46 | 92.59 | 108.80 | 54.52 | 30.85 | 31.44 | 20.73 | 50.20 | **52.71** | 26.60 | 60.37 | 40.93 |
| | Ours-2:8+64LR w/ ALPS | **70.03** | **75.20** | **99.44** | **57.34** | **32.37** | **32.49** | 21.59 | **52.72** | **52.71** | **27.00** | **62.72** | **42.37** |
| | OATS-3:8+64LR | 48.21 | 35.65 | 56.52 | 65.23 | 42.05 | 47.01 | 25.94 | 58.01 | 52.71 | 27.40 | 67.89 | 48.28 |
| | Ours-3:8+64LR w/ SparseGPT | 28.88 | 21.48 | 32.54 | 68.99 | 52.19 | 50.55 | 29.86 | 62.90 | 53.07 | 29.80 | **72.84** | 52.53 |
| | Ours-3:8+64LR w/ ALPS | **25.60** | **19.42** | **27.72** | **69.48** | **54.58** | **53.83** | 30.38 | **65.82** | **54.87** | **33.60** | 70.24 | **54.10** |
| | OATS-4:8+64LR | 15.97 | 10.52 | 16.71 | 75.14 | 68.69 | 66.67 | 40.87 | 69.69 | 54.87 | 39.40 | **79.76** | 61.89 |
| | Ours-4:8+64LR w/ SparseGPT | 14.67 | 9.93 | 15.28 | 76.39 | 70.48 | 68.48 | 42.58 | 70.32 | 54.15 | 39.80 | 79.48 | 62.71 |
| | Ours-4:8+64LR w/ ALPS | **14.14** | **9.58** | **14.78** | **76.93** | **71.35** | **69.15** | **44.45** | **71.74** | **58.48** | **41.40** | 79.69 | **64.15** |
| | OATS-2:4+64LR | 21.05 | 14.42 | 22.62 | 72.85 | 62.47 | 60.69 | 36.35 | 67.09 | 54.87 | 35.00 | 75.11 | 58.05 |
| | Ours-2:4+64LR w/ SparseGPT | 18.06 | 12.66 | 18.66 | 74.86 | 64.77 | **63.85** | 37.37 | 69.22 | 56.68 | 36.40 | 76.12 | 59.91 |
| | Ours-2:4+64LR w/ ALPS | **16.76** | **11.83** | **17.76** | **75.08** | **66.37** | 63.64 | **37.54** | **69.69** | 64.62 | **37.20** | **77.89** | **61.50** |
| | dense | 9.44 | 6.14 | 11.18 | 80.79 | 79.17 | 77.69 | 53.33 | 72.85 | 69.68 | 45.00 | 81.44 | 69.99 |
| Llama3.2-1B | OATS-2:8+64LR | 740.37 | 825.40 | 754.22 | 52.12 | 27.46 | 28.37 | **23.72** | 48.86 | 52.71 | 24.60 | 37.77 | 36.95 |
| | Ours-2:8+64LR w/ SparseGPT | 167.87 | 133.01 | 162.73 | 54.30 | 28.73 | 30.35 | 21.93 | 50.51 | **53.43** | 25.20 | 51.68 | 39.52 |
| | Ours-2:8+64LR w/ ALPS | **125.47** | **114.49** | **136.10** | **55.28** | **29.07** | **32.11** | 20.90 | **51.46** | 52.71 | **25.60** | **55.87** | **40.38** |
| | OATS-3:8+64LR | 96.32 | 74.10 | 93.70 | 59.52 | 33.51 | 36.41 | 22.70 | 50.99 | **52.71** | 25.80 | 62.14 | 42.97 |
| | Ours-3:8+64LR w/ SparseGPT | 45.79 | 34.15 | 52.20 | 62.08 | 38.24 | 41.04 | 23.63 | 54.54 | **52.71** | **30.40** | **62.20** | 45.60 |
| | Ours-3:8+64LR w/ ALPS | **40.95** | **30.33** | **44.58** | **63.76** | **39.64** | **43.14** | **24.66** | **54.93** | **52.71** | 28.20 | 61.65 | **46.09** |
| | OATS-4:8+64LR | 26.75 | 18.49 | 31.94 | 67.30 | 49.52 | 50.51 | 28.41 | 56.67 | **55.96** | 32.40 | 62.87 | 50.46 |
| | Ours-4:8+64LR w/ SparseGPT | 22.71 | 16.05 | 26.80 | 68.28 | 51.42 | 51.22 | 29.18 | **58.64** | 53.07 | 30.00 | 62.51 | 50.54 |
| | Ours-4:8+64LR w/ ALPS | **21.18** | **15.11** | **24.68** | **70.08** | **52.73** | **52.27** | **29.69** | 58.09 | 51.99 | 32.00 | **63.12** | **51.25** |
| | OATS-2:4+64LR | 36.89 | 26.26 | 42.35 | 64.36 | 43.35 | 47.77 | 26.45 | 55.80 | **52.71** | 30.40 | **62.66** | 47.94 |
| | Ours-2:4+64LR w/ SparseGPT | 27.09 | 19.57 | 31.73 | 67.03 | 47.53 | 47.43 | 28.16 | 56.41 | **52.71** | 30.60 | 62.60 | 49.34 |
| | Ours-2:4+64LR w/ ALPS | **25.56** | **18.38** | **29.97** | **68.12** | **47.84** | **48.95** | 28.07 | **59.12** | **52.71** | **32.80** | 62.20 | **49.98** |
| | dense | 14.01 | 9.75 | 17.59 | 74.59 | 63.66 | 60.48 | 36.26 | 60.69 | 56.68 | 37.20 | 63.98 | 56.69 |
| Llama3.2-3B | OATS-2:8+64LR | 444.37 | 543.53 | 851.16 | 52.56 | 27.54 | 27.99 | **23.46** | 50.43 | 51.99 | 26.60 | 37.86 | 37.30 |
| | Ours-2:8+64LR w/ SparseGPT | 122.14 | 114.74 | 165.78 | 54.57 | 28.93 | 30.09 | 21.08 | 49.49 | **52.71** | 26.20 | **62.14** | 40.65 |
| | Ours-2:8+64LR w/ ALPS | **87.85** | **85.97** | **128.34** | **55.77** | **30.33** | **32.24** | 19.80 | **49.80** | **52.71** | 26.60 | 61.31 | **41.07** |
| | OATS-3:8+64LR | 56.80 | 41.62 | 72.75 | 62.68 | 40.49 | 41.84 | 24.06 | 53.91 | 52.35 | 26.60 | 64.10 | 45.75 |
| | Ours-3:8+64LR w/ SparseGPT | 35.07 | 27.12 | 39.63 | 66.43 | 46.08 | 46.42 | 26.62 | **58.17** | **55.96** | 29.00 | 65.47 | 49.27 |
| | Ours-3:8+64LR w/ ALPS | **29.74** | **22.90** | **35.73** | **67.41** | **48.26** | **52.69** | **28.41** | **58.17** | 52.71 | **30.00** | **69.24** | **50.86** |
| | OATS-4:8+64LR w/ ALPS | 18.52 | 12.85 | 20.69 | 72.85 | 61.68 | 62.42 | 36.01 | 64.17 | **60.29** | 36.40 | 72.75 | 58.32 |
| | Ours-4:8+64LR w/ SparseGPT | 17.19 | 12.15 | 19.24 | **73.99** | 63.59 | **62.92** | 36.26 | 67.48 | 57.76 | **39.20** | 71.90 | **59.14** |
| | Ours-4:8+64LR w/ ALPS | **16.40** | **11.62** | **18.17** | 73.83 | **64.24** | 62.29 | 36.26 | **65.19** | 55.96 | 37.00 | **72.87** | 58.46 |
| | OATS-2:4+64LR | 24.32 | 17.06 | 28.54 | 71.98 | 55.87 | 58.80 | 33.36 | 59.91 | 53.07 | 33.80 | 70.18 | 54.62 |
| | Ours-2:4+64LR w/ SparseGPT | 20.82 | 15.65 | 23.77 | 71.71 | 57.88 | 58.84 | **34.39** | 62.12 | **58.12** | 33.60 | 67.92 | 55.57 |
| | Ours-2:4+64LR w/ ALPS | **19.34** | **14.25** | **21.64** | **72.52** | **59.28** | **60.27** | 33.36 | **63.85** | 57.04 | **36.40** | **72.23** | **56.87** |
| | dense | 11.33 | 7.81 | 13.53 | 77.48 | 73.61 | 71.63 | 45.99 | 69.85 | 54.51 | 43.00 | 73.39 | 63.68 |

Table 3: Performance analysis for one-shot N:M sparse plus a 64-rank low-rank matrix decomposition of Llama3 and Llama3.2 models. The rank of the low-rank component is fixed to be $r = 64$. For Perplexity, (↓) lower values are preferred. For zero-shot tasks, (↑) higher values are preferred.

employs an alternating minimization approach to get good solutions. It scales to models with billions of parameters and it is made efficient by exploiting the problem structure (e.g. Hessian-invariance throughout iterations and diagonal rescaling of a minimization approach). Our experiments show that HASSLE-free outperforms existing methods for sparse plus low-rank decomposition of LLMs on a wide-range of LLM evaluation benchmarks. There are many directions for future work. Can we design more efficient algorithms for the decomposition? Extending (**P1**) (subproblem pertaining to sparsity) to include quantization and quantized-sparse compression would be interesting—this would give a better understanding of optimization-based approaches in decomposing dense pre-trained weights into a compressed version (e.g. quantized) plus a low-rank component.

# 7. Acknowledgements

This research is supported in part by grants from Google and the Office of Naval Research. We acknowledge MIT SuperCloud [60] for providing HPC resources that have contributed to the research results reported within this paper. We also acknowledge Google Cloud Credits for computing. The research started when Kayhan Behdin was a graduate student at MIT.

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

# A. Experimental Details

## A.1. Experimental Setup

Following the framework proposed by Frantar and Alistarh [19] for one-shot pruning, we minimize Equation (1) sequentially, layer by layer. For a given layer $\ell$, the input activation matrix $\mathbf{X}$ introduced in Section 3 is the output of the previous $\ell - 1$ compressed layers (sparse plus low-rank) using $N$ calibration samples.

**Implementation details.**

- For the construction of the Hessian matrix $\mathbf{H} = \mathbf{X}^\top \mathbf{X}$ introduced in Section 4, we use the same setup of SparseGPT [19] and we use the author's implementation of SparseGPT—as a pruning plug-in method to minimize (**P1**) (codes available on GitHub).
- We utilize the author's implementation of OATS [33] with the default hyperparameter settings to show LLM evaluation benchmarks and layer-wise reconstruction error in Figure 1.
- The LLM evaluation benchmarks reported in Table 2 are retrieved from the paper ALPS by Meng et al. [18] which uses the same evaluation strategy (and code) we do for the reported tasks [other zero-shot tasks are not reported in ALPS]. We report all zero-shot tasks results for OATS and `HASSLE-free` in Table 4.

## A.2. Hyperparameter Choice

The hyperparameters used in `HASSLE-free` for all experiments and models are the following: $\lambda = 0.01 \, \mathrm{Tr}\,(\mathbf{H})$. $T_{\mathrm{AM}}$ is set to be 80; default value in OATS. $T_{\mathrm{LR}} = 50$; we propose this default value for all experiments. $\eta = 1e^{-2}$; we propose this default value for all experiments (only works well with the scaling introduced in Section 4.2.3). $r$ is either set to 64 and fixed for all layers, or is flexible and given by the formula $r = \left\lfloor (1 - \rho - \frac{N}{M}) \cdot (N_{\mathrm{out}} \cdot N_{\mathrm{in}}) / (N_{\mathrm{out}} + N_{\mathrm{in}}) \right\rfloor$ introduced in Section 5. `Prune`; we propose by default to use SparseGPT. `Optimizer`; we propose the Adam optimizer. **is_scaled**; we propose to set this to True by default. It converges faster in practice and allows to skip the tuning of the learning rate $\eta$.

## A.3. Additional Experimental Resuls

**N:M Sparsity + Fixed Compression Ratio:** This is the same setting described in Section 5. We extend the results reported in Table 2 to include the 8 zero-shot tasks and the Llama3.2 model. Results are reported in Table 4.

**Unstructured Sparsity + Fixed Rank Ratio:** This is the setting introduced in OATS [33]. This scheme takes as inputs a compression ratio $\rho$ (e.g. $50\%$) and rank ratio $\kappa$ (e.g. $0.3$; default value in OATS for the Llama3-8B model). The rank of the low-rank component $r$ and the number of non-zeros $k$ in the unstructured sparsity are given by.

$$r = \left\lfloor \kappa \cdot (1 - \rho) \cdot \frac{N_{\mathrm{out}} \cdot N_{\mathrm{in}}}{N_{\mathrm{out}} + N_{\mathrm{in}}} \right\rfloor, \qquad k = \left\lfloor (1 - \kappa) \cdot (1 - \rho) \cdot N_{\mathrm{out}} \cdot N_{\mathrm{in}} \right\rfloor.$$

See OATS for a discussion on how to choose the rank ratio $\kappa$ for a given model. Note that OATS introduces OWL ratios–different sparsity budgets for different layers to reduce the utility drop. The results for this setting do NOT apply OWL and consider uniform unstructured sparsity throughout layers. The results for OATS and `HASSLE-free` are reported in Table 5.

| Model | Algorithm | Perplexity (↓) | | | Zero-shot (↑) | | | | | | | | |
|---|---|---|---|---|---|---|---|---|---|---|---|---|---|
| | | C4 | WT2 | PTB | PIQA | HS | ARC-E | ARC-C | WG | RTE | OQA | BoolQ | Avg |
| Llama3-8B | OATS-2:8+LR | 21.03 | 14.54 | 24.15 | 73.67 | 62.42 | 59.68 | 37.12 | 65.43 | 55.23 | 36.40 | 73.98 | 57.99 |
| | Ours-2:8+LR w/ SparseGPT | 20.05 | 15.03 | 22.01 | 74.05 | 60.69 | 60.52 | 36.18 | 66.77 | 57.04 | 35.00 | 76.02 | 58.28 |
| | Ours-2:8+LR w/ ALPS | **17.89** | **13.07** | **19.11** | **74.54** | 59.57 | **65.53** | **39.08** | **69.14** | **59.57** | **37.60** | **76.85** | **60.85** |
| | OATS-3:8+LR | 16.87 | 11.43 | 18.53 | 75.24 | 66.90 | 65.91 | 39.85 | 68.90 | 61.37 | 39.00 | 76.61 | 61.72 |
| | Ours-3:8+LR w/ SparseGPT | 16.16 | 11.36 | 16.71 | 75.79 | 67.33 | 67.55 | 41.04 | 69.53 | 58.48 | 39.20 | **79.91** | 62.35 |
| | Ours-3:8+LR w/ ALPS | **14.85** | **10.20** | **15.42** | **77.15** | **69.66** | **69.40** | **43.86** | **70.24** | **63.54** | **39.40** | 77.89 | **63.89** |
| | dense | 9.44 | 6.14 | 11.18 | 80.79 | 79.17 | 77.69 | 53.33 | 72.85 | 69.68 | 45.00 | 81.44 | 69.99 |
| Llama3.2-1B | OATS-2:8+LR | 78.18 | 53.05 | 80.17 | 59.03 | 36.42 | 37.08 | 22.87 | 52.80 | 52.71 | 27.40 | 61.77 | 43.76 |
| | Ours-2:8+LR w/ SparseGPT | 41.08 | 30.92 | 48.85 | 63.22 | 39.07 | 42.55 | **25.77** | 55.17 | **53.07** | 28.00 | **62.11** | 46.12 |
| | Ours-2:8+LR w/ ALPS | **36.29** | **27.35** | **42.07** | **64.09** | **41.55** | **43.64** | 24.91 | **55.88** | **53.07** | **31.00** | 61.90 | **47.01** |
| | OATS-3:8+LR | 42.81 | 29.35 | 47.58 | 63.49 | 42.25 | 43.43 | 25.09 | 54.85 | 52.35 | 29.60 | 62.05 | 46.64 |
| | Ours-3:8+LR w/ SparseGPT | 31.35 | 22.89 | 34.99 | 66.43 | 45.00 | 46.42 | 25.85 | 56.43 | 52.71 | 28.80 | 62.26 | 47.99 |
| | Ours-3:8+LR w/ ALPS | **26.78** | **19.37** | **31.55** | **67.30** | **47.09** | **46.89** | **27.90** | **56.59** | 52.71 | **30.60** | **63.61** | **49.09** |
| | dense | 14.01 | 9.75 | 17.59 | 74.59 | 63.66 | 60.48 | 36.26 | 60.69 | 56.68 | 37.20 | 63.98 | 56.69 |
| Llama3.2-3B | OATS-2:8+LR | 30.73 | 22.65 | 36.31 | 68.55 | 51.76 | 54.46 | 31.14 | 61.17 | **58.48** | 30.80 | **70.43** | 53.35 |
| | Ours-2:8+LR w/ SparseGPT | 25.22 | 19.61 | 29.54 | 69.59 | 52.94 | 55.30 | 29.69 | 62.67 | 55.23 | 30.60 | 69.24 | 53.16 |
| | Ours-2:8+LR w/ ALPS | **22.62** | **17.31** | **26.58** | **70.84** | **55.74** | **57.28** | **32.59** | **64.72** | 54.15 | **35.40** | 67.83 | **54.82** |
| | OATS-3:8+LR | 21.96 | 15.84 | 26.22 | **72.69** | 58.61 | 58.92 | 34.13 | 63.14 | 58.12 | 33.60 | 67.22 | 55.80 |
| | Ours-3:8+LR w/ SparseGPT | 20.03 | 14.85 | 22.92 | 72.42 | 58.92 | 56.69 | 33.53 | 64.01 | 56.32 | **37.00** | 70.31 | 56.15 |
| | Ours-3:8+LR w/ ALPS | **18.21** | **13.50** | **20.96** | **72.96** | **61.62** | **62.25** | **34.98** | **66.38** | **58.48** | 35.20 | 70.12 | **57.75** |
| | dense | 11.33 | 7.81 | 13.53 | 77.48 | 73.61 | 71.63 | 45.99 | 69.85 | 54.51 | 43.00 | 73.39 | 63.68 |

Table 4: Performance analysis for one-shot N:M sparse plus a low-rank matrix decomposition of Llama3 and Llama3.2 models. The compression ratio is fixed to be $\rho = 0.5$. For Perplexity, (↓) lower values are preferred. For zero-shot tasks, (↑) higher values are preferred.

| Model | Algorithm | Perplexity (↓) | | | Zero-shot (↑) | | | | | | | | |
|---|---|---|---|---|---|---|---|---|---|---|---|---|---|
| | | C4 | WT2 | PTB | PIQA | HS | ARC-E | ARC-C | WG | RTE | OQA | BoolQ | Avg |
| Llama3-8B | OATS-60%+LR | 23.61 | 16.52 | 25.85 | 72.91 | 59.65 | 60.10 | 33.36 | 65.35 | 53.07 | 31.60 | 75.96 | 56.50 |
| | Ours-60%+LR w/ SparseGPT | 20.70 | 15.66 | 23.31 | 73.29 | 60.58 | 59.26 | 34.64 | **67.88** | 53.43 | 35.40 | 75.08 | 57.44 |
| | Ours-60%+LR w/ ALPS | **18.55** | **13.69** | **20.58** | **73.88** | **63.75** | **63.38** | **36.77** | 67.25 | **58.48** | **37.60** | **76.15** | **59.66** |
| | OATS-70%+LR | 106.98 | 81.77 | 110.44 | 55.60 | 30.30 | 32.45 | 20.05 | 49.96 | 52.71 | 27.00 | 62.35 | 41.30 |
| | Ours-70%+LR w/ SparseGPT | 50.07 | 49.13 | 60.89 | 60.50 | 39.67 | 37.21 | 23.38 | 55.25 | 52.71 | 27.40 | 66.09 | 45.27 |
| | Ours-70%+LR w/ ALPS | **41.50** | **34.38** | **47.66** | **64.58** | **41.76** | **42.30** | **25.77** | **60.62** | 52.71 | **29.80** | **68.35** | **48.24** |
| | OATS-80%+LR | 748.40 | 909.75 | 1601.02 | 52.29 | 27.25 | 26.81 | **24.40** | 47.59 | 52.71 | 26.60 | 37.83 | 36.93 |
| | Ours-80%+LR w/ SparseGPT | 164.27 | 265.28 | 235.38 | 53.32 | 28.53 | 29.38 | 20.22 | 49.49 | 52.71 | 26.60 | 38.84 | 37.39 |
| | Ours-80%+LR w/ ALPS | **120.66** | **150.32** | **148.24** | **53.86** | **29.27** | **29.55** | 21.08 | **50.67** | 52.71 | **27.40** | **47.92** | **39.06** |
| | dense | 11.33 | 7.81 | 13.53 | 77.48 | 73.61 | 71.63 | 45.99 | 69.85 | 54.51 | 43.00 | 73.39 | 63.68 |
| Llama3.2-1B | OATS-60%+LR | 73.87 | 54.42 | 79.95 | 58.49 | 35.02 | 35.35 | 22.95 | 50.83 | 52.71 | 26.20 | **62.20** | 42.97 |
| | Ours-60%+LR w/ SparseGPT | 53.56 | 41.63 | 58.35 | 62.68 | 38.62 | 39.69 | 25.34 | **55.56** | 52.71 | **29.00** | 62.08 | 45.71 |
| | Ours-60%+LR w/ ALPS | **44.60** | **36.72** | **46.82** | **64.47** | **40.37** | **42.17** | **25.51** | 54.78 | 52.71 | 26.40 | **62.20** | **46.08** |
| | OATS-70%+LR | 326.24 | 311.00 | 315.90 | 54.41 | 28.66 | 29.25 | **23.29** | 51.07 | 52.71 | **26.60** | 59.82 | 40.73 |
| | Ours-70%+LR w/ SparseGPT | 156.56 | 135.92 | 153.59 | 55.60 | 29.31 | 31.48 | 20.90 | 50.99 | 52.71 | 26.40 | **62.14** | 41.19 |
| | Ours-70%+LR w/ ALPS | **97.60** | **83.34** | **98.42** | **56.80** | **30.71** | **34.13** | 20.56 | **53.35** | 52.71 | 25.00 | 61.28 | **41.82** |
| | OATS-80%+LR | 1856.90 | 3129.91 | 4402.58 | 50.71 | 26.20 | 26.26 | **24.15** | 50.04 | 52.71 | 25.80 | 37.83 | 36.71 |
| | Ours-80%+LR w/ SparseGPT | 341.34 | 378.74 | 400.86 | 52.77 | 26.87 | **28.96** | 21.76 | 50.75 | 53.07 | 25.60 | 39.91 | 37.46 |
| | Ours-80%+LR w/ ALPS | **286.65** | **269.78** | **262.04** | **53.26** | **27.59** | 28.70 | 21.67 | 48.70 | 53.07 | **26.00** | **42.08** | **37.64** |
| | dense | 11.33 | 7.81 | 13.53 | 77.48 | 73.61 | 71.63 | 45.99 | 69.85 | 54.51 | 43.00 | 73.39 | 63.68 |
| Llama3.2-3B | OATS-60%+LR | 34.57 | 24.94 | 41.51 | 67.79 | 48.40 | 52.57 | 30.38 | 57.70 | 54.15 | 30.80 | 65.66 | 50.93 |
| | Ours-60%+LR w/ SparseGPT | 27.67 | 21.90 | 33.40 | 69.15 | 52.04 | 51.26 | 29.52 | 61.96 | 58.12 | 29.80 | **69.72** | 52.70 |
| | Ours-60%+LR w/ ALPS | **24.43** | **18.68** | **28.38** | **70.24** | **54.42** | **53.58** | **30.55** | **63.22** | **58.48** | **34.60** | 68.32 | **54.18** |
| | OATS-70%+LR | 155.48 | 121.76 | 167.60 | 54.57 | 29.83 | 30.43 | 21.42 | 49.64 | 52.71 | **28.20** | 60.43 | 40.90 |
| | Ours-70%+LR w/ SparseGPT | 78.65 | 75.23 | 103.10 | 58.43 | 32.44 | 35.27 | 21.67 | 49.41 | 52.71 | 27.00 | **62.29** | 42.40 |
| | Ours-70%+LR w/ ALPS | **56.85** | **50.70** | **74.66** | **60.23** | **36.09** | **37.42** | **22.61** | **53.12** | 52.71 | 28.00 | 62.02 | **44.02** |
| | OATS-80%+LR | 1085.27 | 1610.87 | 2546.29 | 50.60 | 26.60 | 26.68 | **24.40** | 47.67 | 52.71 | **26.60** | 37.83 | 36.64 |
| | Ours-80%+LR w/ SparseGPT | 217.62 | 320.98 | 320.02 | 53.10 | 27.86 | **29.12** | 22.01 | 47.75 | 50.54 | **26.60** | 46.61 | 37.95 |
| | Ours-80%+LR w/ ALPS | **149.85** | **185.15** | **229.48** | **53.37** | **28.60** | 29.00 | 19.71 | **50.43** | 52.71 | 25.80 | **56.73** | **39.54** |
| | dense | 11.33 | 7.81 | 13.53 | 77.48 | 73.61 | 71.63 | 45.99 | 69.85 | 54.51 | 43.00 | 73.39 | 63.68 |

Table 5: Performance analysis for one-shot unstructured sparsity plus a low-rank matrix decomposition of Llama3 and Llama3.2-3B model. The rank ratio of the low-rank component is fixed to be $\kappa = 0.3$. For Perplexity, (↓) lower values are preferred. For zero-shot tasks, (↑) higher values are preferred.

