# OpenReview forum: "A unified framework for Sparse plus Low-Rank Matrix Decomposition for LLMs"
_CPAL.cc/2025/Proceedings_Track — CPAL 2025 (Proceedings Track) Oral_

### Official Review · Reviewer_V8Qk · 2024-12-25
**Well motivated framework for compression, relatively weak methodology.**

**Rating:** 8
**Confidence:** 3

**Review:**

### Summary:
This paper introduces **HASSLE-free**, a unified framework for layer-wise compression of LLMs. It achieves an **exact** sparse plus low-rank decomposition of weights, supporting diverse sparse patterns and rank constraints.

To construct this framework, the authors decompose the approximation problem into two minimization subproblems using alternating optimization. They explore three common formulations: data-free, diagonal approximation, and regularized full Hessian, drawing connections to popular methods like Wanda and OATS. Leveraging SparseGPT for sparse minimization and low-rank reparameterization for low-rank minimization, the authors demonstrate superior performance compared to SoTA pruning methods, including OATS, Wanda, and DSNoT.

### Strengths:
1) Writing and clarity: The paper is well-written and logically structured, offering an intuitive deconstruction of the sparse + low-rank approximation problem. The detailed derivations are rigorous and accessible, making the paper a pleasure to read.
2) Comprehensive experiments: The experiments cover SoTA open-source LLMs (e.g. Llama-3, Llama-3.2), benchmarking against strong baselines like SparseGPT, Wanda, and DSNoT under various sparsity and rank configurations.

### Weaknesses:
1) Trade-offs of exact approximation: The utility of solving the exact approximation problem is unclear. While exactness reduces the reconstruction loss (as seen in Figure 1) and shows improvements in Tables 1 and 2 in the paper, the downstream performance impact is not **generally** guaranteed to surpass approximate methods, especially when given fixed complexity constraints.
2) Computational overhead: Exact approximation involves increased computational costs for computing and inverting the Hessian X^TX. The paper does not provide efficiency metrics, such as runtime comparisons or complexity analysis for the proposed approach.

### Questions:
1) Can the authors compare computational complexity or compression speed with baseline methods?
2) Can the framework incorporate an approximation module early in the algorithm to allow compatibility with surrogate Hessians, potentially reducing computational costs?

A minor catch is that in sec. 4.1.2; it should be $S_{ij} = |\tilde W_{ij}| \|X_j\|_2= |\tilde{W} D|_{ij}$?

---

### Official Review · Reviewer_YnTi · 2025-01-08
**LLM Compression: Sparse + Low-rank**

**Rating:** 6
**Confidence:** 4

**Review:**

The paper proposes a method that combines SparseGPT with low-rank factorization for LLM compression, demonstrating superior results across various benchmarks.

**Pros:**
* The paper is well-written and clearly presented overall.
* The paper effectively connects prior work through a unified optimization framework.
* The layer-by-layer pruning strategy is logical and well-motivated, particularly given the impact of earlier layers on subsequent activations. (although I feel like this should be mentioned in the main paper not the appendix).

**Cons:**
* The sparsity constraint is not formally discussed in the paper, while the $C_s$ constraint is mentioned in Line 136, it pertains only to low-rankness. Also the definition of N : M sparsity is not introduced.
* All experiments done in the paper are based on the Llama model, it would be better to include comparisons for more models (not necessarily large models) and other methods (e.g. [1][2]).
* Minor: $T_{LR}$ is listed as a input parameter in Algorithm 1, but the authors write $N$ in the algorithm instead.

[1] Hsu et al; Language model compression with weighted low-rank factorization.

[2] Zhang et al; PLATON: Pruning Large Transformer Models with Upper Confidence Bound of Weight Importance.

---

### Official Review · Reviewer_ajde · 2025-01-08
**Review of sparse + low rank pruning**

**Rating:** 6
**Confidence:** 4

**Review:**

This paper introduces a method to decompose weights into sparse and low rank components for model compression/efficiency.

Pros:
- Interesting method with many experiments to show its better performance over its main baseline, OATS

Cons:
- Missing efficiency evaluations which is a key motivation for structured pruning.
- Contribution seems limited given more adaptive structured pruning methods [1, 2, 3].

Questions:
1) The accuracy and perplexity are still quite far from the dense model. What do you hypothesize is missing? Is there some structure in the residual that wasn't captured?
2) How long did it take to prune a specific model (ballpark is fine)?
3) Why is the focus of the paper on N:M sparsity? Is there something limiting your method's application to other forms of structured sparsity?

[1] Liu et al., Deja Vu: Contextual Sparsity for Efficient LLMs at Inference Time, 2023.

[2] Lee et al., CATS: Contextually-Aware Thresholding for Sparsity in Large Language Models, 2024.

[3] Dong et al., Prompt-prompted Adaptive Structured Pruning for Efficient LLM Generation, 2024.

---

### Meta-Review · Area_Chair_8MG5 · 2025-02-06

**Recommendation:** Accept (Poster)
**Confidence:** 4

**Metareview:**

This paper introduces HASSLE-free, a promising method for layer-wise compression of LLMs through exact sparse and low-rank weight decomposition. The approach is well-explained and supported by comprehensive experiments, demonstrating competitive performance against state-of-the-art methods like OATS, Wanda, and DSNoT. While the method shows superior results in certain benchmarks, the practical applicability of the exact approximation and its computational costs remain unclear. Please address these concerns based on the responses to reviewers.

Based on the above evaluation, the paper is recommended for acceptance.

---

### Decision · Program_Chairs · 2025-02-11

Accept (Oral)